# Projective dictionary pair learning for pattern classification

**Shuhang Gu**[1], **Lei Zhang**[1], **Wangmeng Zuo**[2], **Xiangchu Feng**[3]

[1]Dept. of Computing, The Hong Kong Polytechnic University, Hong Kong, China
[2]School of Computer Science and Technology, Harbin Institute of Technology, Harbin, China
[3]Dept. of Applied Mathematics, Xidian University, Xi′an, China
{cssgu, cslzhang}@comp.polyu.edu.hk
cswmzuo@gmail.com, xcfeng@mail.xidian.edu.cn

## Abstract

Discriminative dictionary learning (DL) has been widely studied in various pattern classification problems. Most of the existing DL methods aim to learn a synthesis dictionary to represent the input signal while enforcing the representation coefficients and/or representation residual to be discriminative. However, the $\ell_0$ or $\ell_1$-norm sparsity constraint on the representation coefficients adopted in most DL methods makes the training and testing phases time consuming. We propose a new discriminative DL framework, namely projective dictionary pair learning (DPL), which learns a synthesis dictionary and an analysis dictionary jointly to achieve the goal of signal representation and discrimination. Compared with conventional DL methods, the proposed DPL method can not only greatly reduce the time complexity in the training and testing phases, but also lead to very competitive accuracies in a variety of visual classification tasks.

## 1 Introduction

Sparse representation represents a signal as the linear combination of a small number of atoms chosen out of a dictionary, and it has achieved a big success in various image processing and computer vision applications [1, 2]. The dictionary plays an important role in the signal representation process [3]. By using a predefined analytical dictionary (e.g., wavelet dictionary, Gabor dictionary) to represent a signal, the representation coefficients can be produced by simple inner product operations. Such a fast and explicit coding makes analytical dictionary very attractive in image representation; however, it is less effective to model the complex local structures of natural images.

Sparse representation with a synthesis dictionary has been widely studied in recent years [2, 4, 5]. With synthesis dictionary, the representation coefficients of a signal are usually obtained via an $\ell_p$-norm ($p \leq 1$) sparse coding process, which is computationally more expensive than analytical dictionary based representation. However, synthesis based sparse representation can better model the complex image local structures and it has led to many state-of-the-art results in image restoration [6]. Another important advantage lies in that the synthesis based sparse representation model allows us to easily learn a desired dictionary from the training data. The seminal work of KSVD [1] tells us that an over-complete dictionary can be learned from example natural images, and it can lead to much better image reconstruction results than the analytically designed off-the-shelf dictionaries. Inspired by KSVD, many dictionary learning (DL) methods have been proposed and achieved state-of-the-art performance in image restoration tasks.

The success of DL in image restoration problems triggers its applications in image classification tasks. Different from image restoration, assigning the correct class label to the test sample is the goal of classification problems; therefore, the discrimination capability of the learned dictionary is

of the major concern. To this end, supervised dictionary learning methods have been proposed to promote the discriminative power of the learned dictionary [4, 5, 7, 8, 9]. By encoding the query sample over the learned dictionary, both the coding coefficients and the coding residual can be used for classification, depending on the employed DL model. Discriminative DL has led to many state-of-the-art results in pattern recognition problems.

One popular strategy of discriminative DL is to learn a shared dictionary for all classes while enforcing the coding coefficients to be discriminative [4, 5, 7]. A classifier on the coding coefficients can be trained simultaneously to perform classification. Mairal et al. [7] proposed to learn a dictionary and a corresponding linear classifier in the coding vector space. In the label consistent KSVD (LC-KSVD) method, Jiang et al. [5] introduced a binary class label sparse code matrix to encourage samples from the same class to have similar sparse codes. In [4], Mairal et al. proposed a task driven dictionary learning (TDDL) framework, which minimizes different risk functions of the coding coefficients for different tasks.

Another popular line of research in DL attempts to learn a structured dictionary to promote discrimination between classes [2, 8, 9, 10]. The atoms in the structured dictionary have class labels, and the class-specific representation residual can be computed for classification. Ramirez et al. [8] introduced an incoherence promotion term to encourage the sub-dictionaries of different classes to be independent. Yang et al. [9] proposed a Fisher discrimination dictionary learning (FDDL) method which applies the Fisher criterion to both representation residual and representation coefficient. Wang et al. [10] proposed a max-margin dictionary learning (MMDL) algorithm from the large margin perspective.

In most of the existing DL methods, $\ell_0$-norm or $\ell_1$-norm is used to regularize the representation coefficients since sparser coefficients are more likely to produce better classification results. Hence a sparse coding step is generally involved in the iterative DL process. Although numerous algorithms have been proposed to improve the efficiency of sparse coding [11, 12], the use of $\ell_0$-norm or $\ell_1$-norm sparsity regularization is still a big computation burden and makes the training and testing inefficient.

It is interesting to investigate whether we can learn discriminative dictionaries but without the costly $\ell_0$-norm or $\ell_1$-norm sparsity regularization. In particular, it would be very attractive if the representation coefficients can be obtained by linear projection instead of nonlinear sparse coding. To this end, in this paper we propose a projective dictionary pair learning (DPL) framework to learn a synthesis dictionary and an analysis dictionary jointly for pattern classification. The analysis dictionary is trained to generate discriminative codes by efficient linear projection, while the synthesis dictionary is trained to achieve class-specific discriminative reconstruction. The idea of using functions to predict the representation coefficients is not new, and fast approximate sparse coding methods have been proposed to train nonlinear functions to generate sparse codes [13, 14]. However, there are clear difference between our DPL model and these methods. First, in DPL the synthesis dictionary and analysis dictionary are trained jointly, which ensures that the representation coefficients can be approximated by a simple linear projection function. Second, DPL utilizes class label information and promotes discriminative power of the representation codes.

One related work to this paper is the analysis-based sparse representation prior learning [15, 16], which represents a signal from a dual viewpoint of the commonly used synthesis model. Analysis prior learning tries to learn a group of analysis operators which have sparse responses to the latent clean signal. Sprechmann et al. [17] proposed to train a group of analysis operators for classification; however, in the testing phase a costly sparsity-constrained optimization problem is still required. Feng et al. [18] jointly trained a dimensionality reduction transform and a dictionary for face recognition. The discriminative dictionary is trained in the transformed space, and sparse coding is needed in both the training and testing phases.

The contribution of our work is two-fold. First, we introduce a new DL framework, which extends the conventional discriminative synthesis dictionary learning to discriminative synthesis and analysis dictionary pair learning (DPL). Second, the DPL utilizes an analytical coding mechanism and it largely improves the efficiency in both the training and testing phases. Our experiments in various visual classification datasets show that DPL achieves very competitive accuracy with state-of-the-art DL algorithms, while it is significantly faster in both training and testing.

## 2 Projective Dictionary Pair Learning

### 2.1 Discriminative dictionary learning

Denote by $X = [X_1, \ldots, X_k, \ldots, X_K]$ a set of $p$-dimensional training samples from $K$ classes, where $X_k \in \Re^{p \times n}$ is the training sample set of class $k$, and $n$ is the number of samples of each class. Discriminative DL methods aim to learn an effective data representation model from $X$ for classification tasks by exploiting the class label information of training data. Most of the state-of-the-art discriminative DL methods [5, 7, 9] can be formulated under the following framework:

$$\min_{D,A} \parallel X - DA \parallel_F^2 + \lambda \parallel A \parallel_p + \Psi(D, A, Y), \tag{1}$$

where $\lambda \geq 0$ is a scalar constant, $Y$ represents the class label matrix of samples in $X$, $D$ is the synthesis dictionary to be learned, and $A$ is the coding coefficient matrix of $X$ over $D$. In the training model (1), the data fidelity term $\parallel X - DA \parallel_F^2$ ensures the representation ability of $D$; $\parallel A \parallel_p$ is the $\ell_p$-norm regularizer on $A$; and $\Psi(D, A, Y)$ stands for some discrimination promotion function, which ensures the discrimination power of $D$ and $A$.

As we introduced in Section 1, some DL methods [4, 5, 7] learn a shared dictionary for all classes and a classifier on the coding coefficients simultaneously, while some DL methods [8, 9, 10] learn a structured dictionary to promote discrimination between classes. However, they all employ $\ell_0$ or $\ell_1$-norm sparsity regularizer on the coding coefficients, making the training stage and the consequent testing stage inefficient.

In this work, we extend the conventional DL model in (1), which learns a discriminative synthesis dictionary, to a novel DPL model, which learns a pair of synthesis and analysis dictionaries. No costly $\ell_0$ or $\ell_1$-norm sparsity regularizer is required in the proposed DPL model, and the coding coefficients can be explicitly obtained by linear projection. Fortunately, DPL does not sacrifice the classification accuracy while achieving significant improvement in the efficiency, as demonstrated by our extensive experiments in Section 3.

### 2.2 The dictionary pair learning model

The conventional discriminative DL model in (1) aims to learn a synthesis dictionary $D$ to sparsely represent the signal $X$, and a costly $\ell_1$-norm sparse coding process is needed to resolve the code $A$. Suppose that if we can find an analysis dictionary, denoted by $P \in \Re^{mK \times p}$, such that the code $A$ can be analytically obtained as $A = PX$, then the representation of $X$ would become very efficient. Based on this idea, we propose to learn such an analysis dictionary $P$ together with the synthesis dictionary $D$, leading to the following DPL model:

$$\{P^*, D^*\} = \arg\min_{P,D} \parallel X - DPX \parallel_F^2 + \Psi(D, P, X, Y), \tag{2}$$

where $\Psi(D, P, X, Y)$ is some discrimination function. $D$ and $P$ form a dictionary pair: the analysis dictionary $P$ is used to analytically code $X$, and the synthesis dictionary $D$ is used to reconstruct $X$.

The discrimination power of the DPL model depends on the suitable design of $\Psi(D, P, X, Y)$. We propose to learn a structured synthesis dictionary $D = [D_1, \ldots, D_k, \ldots, D_K]$ and a structured analysis dictionary $P = [P_1; \ldots; P_k; \ldots; P_K]$, where $\{D_k \in \Re^{p \times m}, P_k \in \Re^{m \times p}\}$ forms a sub-dictionary pair corresponding to class $k$. Recent studies on sparse subspace clustering [19] have proved that a sample can be represented by its corresponding dictionary if the signals satisfy certain incoherence condition. With the structured analysis dictionary $P$, we want that the sub-dictionary $P_k$ can project the samples from class $i, i \neq k$, to a nearly null space, i.e.,

$$P_k X_i \approx 0, \forall k \neq i. \tag{3}$$

Clearly, with (3) the coefficient matrix $PX$ will be nearly block diagonal. On the other hand, with the structured synthesis dictionary $D$, we want that the sub-dictionary $D_k$ can well reconstruct the data matrix $X_k$ from its projective code matrix $P_k X_k$; that is, the dictionary pair should minimize the reconstruction error:

$$\min_{P,D} \sum_{k=1}^{K} \parallel X_k - D_k P_k X_k \parallel_F^2. \tag{4}$$

Based on the above analysis, we can readily have the following DPL model:

$$\{P^*, D^*\} = \arg\min_{P,D} \sum_{k=1}^{K} \parallel X_k - D_k P_k X_k \parallel_F^2 + \lambda \parallel P_k \bar{X}_k \parallel_F^2, \quad s.t. \parallel d_i \parallel_2^2 \leq 1. \tag{5}$$

---

**Algorithm 1** Discriminative synthesis&analysis dictionary pair learning (DPL)

---

**Input:** Training samples for $K$ classes $\boldsymbol{X} = [\boldsymbol{X}_1, \boldsymbol{X}_2, \ldots, \boldsymbol{X}_K]$, parameter $\lambda, \tau, m$;
 1: Initialize $\boldsymbol{D}^{(0)}$ and $\boldsymbol{P}^{(0)}$ as random matrixes with unit Frobenious norm, $t = 0$;
 2: **while** not converge **do**
 3:    $t \leftarrow t + 1$;
 4:    **for** $i$=1:$K$ **do**
 5:       Update $\boldsymbol{A}_k^{(t)}$ by (8);
 6:       Update $\boldsymbol{P}_k^{(t)}$ by (10);
 7:       Update $\boldsymbol{D}_k^{(t)}$ by (12);
 8:    **end for**
 9: **end while**
**Output:** Analysis dictionary $\boldsymbol{P}$, synthesis dictionary $\boldsymbol{D}$.

---

where $\bar{\boldsymbol{X}}_k$ denotes the complementary data matrix of $\boldsymbol{X}_k$ in the whole training set $\boldsymbol{X}$, $\lambda > 0$ is a scalar constant, and $\boldsymbol{d}_i$ denotes the $i$th atom of synthesis dictionary $\boldsymbol{D}$. We constrain the energy of each atom $\boldsymbol{d}_i$ in order to avoid the trivial solution of $\boldsymbol{P}_k = \boldsymbol{0}$ and make the DPL more stable.

The DPL model in (5) is not a sparse representation model, while it enforces group sparsity on the code matrix $\boldsymbol{PX}$ (i.e., $\boldsymbol{PX}$ is nearly block diagonal). Actually, the role of sparse coding in classification is still an open problem, and some researchers argued that sparse coding may not be crucial to classification tasks [20, 21]. Our findings in this work are supportive to this argument. The D-PL model leads to very competitive classification performance with those sparse coding based DL models, but it is much faster.

### 2.3 Optimization

The objective function in (5) is generally non-convex. We introduce a variable matrix $\boldsymbol{A}$ and relax (5) to the following problem:

$$\{\boldsymbol{P}^*, \boldsymbol{A}^*, \boldsymbol{D}^*\} = \arg\min_{\boldsymbol{P},\boldsymbol{A},\boldsymbol{D}} \sum_{k=1}^{K} \|\boldsymbol{X}_i - \boldsymbol{D}_k\boldsymbol{A}_k\|_F^2 + \tau \|\boldsymbol{P}_k\boldsymbol{X}_k - \boldsymbol{A}_k\|_F^2 + \lambda \|\boldsymbol{P}_k\bar{\boldsymbol{X}}_k\|_F^2, \quad s.t. \|\boldsymbol{d}_i\|_2^2 \leq 1. \quad (6)$$

where $\tau$ is a scalar constant. All terms in the above objective function are characterized by Frobenius norm, and (6) can be easily solved. We initialize the analysis dictionary $\boldsymbol{P}$ and synthesis dictionary $\boldsymbol{D}$ as random matrices with unit Frobenius norm, and then alternatively update $\boldsymbol{A}$ and $\{\boldsymbol{D}, \boldsymbol{P}\}$. The minimization can be alternated between the following two steps.

(1) Fix $\boldsymbol{D}$ and $\boldsymbol{P}$, update $\boldsymbol{A}$

$$\boldsymbol{A}^* = \arg\min_{\boldsymbol{A}} \sum_{k=1}^{K} \|\boldsymbol{X}_k - \boldsymbol{D}_k\boldsymbol{A}_k\|_F^2 + \tau \|\boldsymbol{P}_k\boldsymbol{X}_k - \boldsymbol{A}_k\|_F^2. \quad (7)$$

This is a standard least squares problem and we have the closed-form solution:

$$\boldsymbol{A}_k^* = (\boldsymbol{D}_k^T\boldsymbol{D}_k + \tau\boldsymbol{I})^{-1}(\tau\boldsymbol{P}_k\boldsymbol{X}_k + \boldsymbol{D}_k^T\boldsymbol{X}_k). \quad (8)$$

(2) Fix $\boldsymbol{A}$, update $\boldsymbol{D}$ and $\boldsymbol{P}$:

$$\begin{cases} \boldsymbol{P}^* = \arg\min_{\boldsymbol{P}} \sum_{k=1}^{K} \tau \|\boldsymbol{P}_k\boldsymbol{X}_k - \boldsymbol{A}_k\|_F^2 + \lambda \|\boldsymbol{P}_k\bar{\boldsymbol{X}}_k\|_F^2; \\ \boldsymbol{D}^* = \arg\min_{\boldsymbol{D}} \sum_{k=1}^{K} \|\boldsymbol{X}_k - \boldsymbol{D}_k\boldsymbol{A}_k\|_F^2, \quad s.t. \|\boldsymbol{d}_i\|_2^2 \leq 1. \end{cases} \quad (9)$$

The closed-form solutions of $\boldsymbol{P}$ can be obtained as:

$$\boldsymbol{P}_k^* = \tau\boldsymbol{A}_k\boldsymbol{X}_k^T(\tau\boldsymbol{X}_k\boldsymbol{X}_k^T + \lambda\bar{\boldsymbol{X}}_k\bar{\boldsymbol{X}}_k^T + \gamma\boldsymbol{I})^{-1}, \quad (10)$$

where $\gamma = 10e^{-4}$ is a small number. The $\boldsymbol{D}$ problem can be optimized by introducing a variable $\boldsymbol{S}$:

$$\min_{\boldsymbol{D},\boldsymbol{S}} \sum_{k=1}^{K} \|\boldsymbol{X}_k - \boldsymbol{D}_k\boldsymbol{A}_k\|_F^2, \quad s.t. \boldsymbol{D} = \boldsymbol{S}, \quad \|\boldsymbol{s}_i\|_2^2 \leq 1. \quad (11)$$

The optimal solution of (11) can be obtained by the ADMM algorithm:

$$\begin{cases} \boldsymbol{D}^{(r+1)} = \arg\min_{\boldsymbol{D}} \sum_{k=1}^{K} \|\boldsymbol{X}_k - \boldsymbol{D}_k\boldsymbol{A}_k\|_F^2 + \rho \|\boldsymbol{D}_k - \boldsymbol{S}_k^{(r)} + \boldsymbol{T}_k^{(r)}\|_F^2, \\ \boldsymbol{S}^{(r+1)} = \arg\min_{\boldsymbol{S}} \sum_{k=1}^{K} \rho \|\boldsymbol{D}_k^{(r+1)} - \boldsymbol{S}_k + \boldsymbol{T}_k^{(r)}\|_F^2, \quad s.t. \|\boldsymbol{s}_i\|_2^2 \leq 1, \\ \boldsymbol{T}^{(r+1)} = \boldsymbol{T}^{(r)} + \boldsymbol{D}_k^{(r+1)} - \boldsymbol{S}_k^{(r+1)}, \text{ update } \rho \text{ if appropriate.} \end{cases} \quad (12)$$

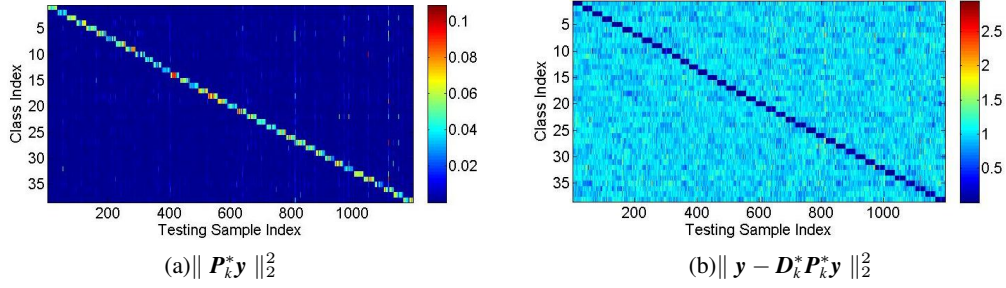

$$(a)\| \boldsymbol{P}_k^*\boldsymbol{y} \|_2^2 \qquad\qquad (b)\| \boldsymbol{y} - \boldsymbol{D}_k^*\boldsymbol{P}_k^*\boldsymbol{y} \|_2^2$$

Figure 1: (a) The representation codes and (b) reconstruction error on the Extended YaleB dataset.

In each step of optimization, we have closed form solutions for variables $\boldsymbol{A}$ and $\boldsymbol{P}$, and the ADMM based optimization of $\boldsymbol{D}$ converges rapidly. The training of the proposed DPL model is much faster than most of previous discriminative DL methods. The proposed DPL algorithm is summarized in Algorithm 1. When the difference between the energy in two adjacent iterations is less than 0.01, the iteration stops. The analysis dictionary $\boldsymbol{P}$ and the synthesis dictionary $\boldsymbol{D}$ are then output for classification.

One can see that the first sub-objective function in (9) is a discriminative analysis dictionary learner, focusing on promoting the discriminative power of $\boldsymbol{P}$; the second sub-objective function in (9) is a representative synthesis dictionary learner, aiming to minimize the reconstruction error of the input signal with the coding coefficients generated by the analysis dictionary $\boldsymbol{P}$. When the minimization process converges, a balance between the discrimination and representation power of the model can be achieved.

## 2.4 Classification scheme

In the DPL model, the analysis sub-dictionary $\boldsymbol{P}_k^*$ is trained to produce small coefficients for samples from classes other than $k$, and it can only generate significant coding coefficients for samples from class $k$. Meanwhile, the synthesis sub-dictionary $\boldsymbol{D}_k^*$ is trained to reconstruct the samples of class $k$ from their projective coefficients $\boldsymbol{P}_k^*\boldsymbol{X}_k$; that is, the residual $\| \boldsymbol{X}_k - \boldsymbol{D}_k^*\boldsymbol{P}_k^*\boldsymbol{X}_k \|_F^2$ will be small. On the other hand, since $\boldsymbol{P}_k^*\boldsymbol{X}_i, i \neq k$, will be small and $\boldsymbol{D}_k^*$ is not trained to reconstruct $\boldsymbol{X}_i$, the residual $\| \boldsymbol{X}_i - \boldsymbol{D}_k^*\boldsymbol{P}_k^*\boldsymbol{X}_i \|_F^2$ will be much larger than $\| \boldsymbol{X}_k - \boldsymbol{D}_k^*\boldsymbol{P}_k^*\boldsymbol{X}_k \|_F^2$.

In the testing phase, if the query sample $\boldsymbol{y}$ is from class $k$, its projective coding vector by $\boldsymbol{P}_k^*$ (i.e., $\boldsymbol{P}_k^*\boldsymbol{y}$ ) will be more likely to be significant, while its projective coding vectors by $\boldsymbol{P}_i^*, i \neq k$, tend to be small. Consequently, the reconstruction residual $\| \boldsymbol{y} - \boldsymbol{D}_k^*\boldsymbol{P}_k^*\boldsymbol{y} \|_2^2$ tends to be much smaller than the residuals $\| \boldsymbol{y} - \boldsymbol{D}_i^*\boldsymbol{P}_i^*\boldsymbol{y} \|_2^2, i \neq k$. Let us use the Extended YaleB face dataset [22] to illustrate this. (The detailed experimental setting can be found in Section 3.) Fig. 1(a) shows the $\ell_2$-norm of the coefficients $\boldsymbol{P}_k^*\boldsymbol{y}$, where the horizontal axis refers to the index of $\boldsymbol{y}$ and the vertical axis refers to the index of $\boldsymbol{P}_k^*$ . One can clearly see that $\| \boldsymbol{P}_k^*\boldsymbol{y} \|_2^2$ has a nearly block diagonal structure, and the diagonal blocks are produced by the query samples which have the same class labels as $\boldsymbol{P}_k^*$ . Fig. 1(b) shows the reconstruction residual $\| \boldsymbol{y} - \boldsymbol{D}_k^*\boldsymbol{P}_k^*\boldsymbol{y} \|_2^2$. One can see that $\| \boldsymbol{y} - \boldsymbol{D}_k^*\boldsymbol{P}_k^*\boldsymbol{y} \|_2^2$ also has a block diagonal structure, and only the diagonal blocks have small residuals. Clearly, the class-specific reconstruction residual can be used to identify the class label of $\boldsymbol{y}$, and we can naturally have the following classifier associated with the DPL model:

$$identity(\boldsymbol{y}) = \arg\min_i \| \boldsymbol{y} - \boldsymbol{D}_i\boldsymbol{P}_i\boldsymbol{y} \|_2 . \tag{13}$$

## 2.5 Complexity and Convergence

**Complexity** In the training phase of DPL, $\boldsymbol{A}_k$, $\boldsymbol{P}_k$ and $\boldsymbol{D}_k$ are updated alternatively. In each iteration, the time complexities of updating $\boldsymbol{A}_k$, $\boldsymbol{P}_k$ and $\boldsymbol{D}_k$ are $\boldsymbol{O}(mpn + m^3 + m^2n)$, $\boldsymbol{O}(mnp + p^3 + mp^2)$ and $\boldsymbol{O}(W(pmn + m^3 + m^2p + p^2m))$, respectively, where $W$ is the iteration number in ADMM algorithm for updating $\boldsymbol{D}$. We experimentally found that in most cases $W$ is less than 20. In many applications, the number of training samples and the number of dictionary atoms for each class are much smaller than the dimension $p$. Thus the major computational burden in the training phase of DPL is on updating $\boldsymbol{P}_k$, which involves an inverse of a $p \times p$ matrix $\{\tau\boldsymbol{X}_k\boldsymbol{X}_k^T + \lambda\bar{\boldsymbol{X}}_k\bar{\boldsymbol{X}}_k^T + \gamma\boldsymbol{I}\}$. Fortunately, this

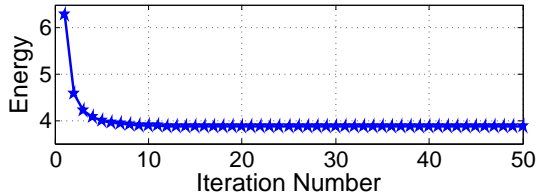

Figure 2: The convergence curve of DPL on the AR database.

matrix will not change in the iteration, and thus the inverse of it can be pre-computed. This greatly accelerates the training process.

In the testing phase, our classification scheme is very efficient. The computation of class-specific reconstruction error $\parallel \boldsymbol{y} - \boldsymbol{D}_k^* \boldsymbol{P}_k^* \boldsymbol{y} \parallel_2$ only has a complexity of $\boldsymbol{O}(mp)$. Thus, the total complexity of our model to classify a query sample is $\boldsymbol{O}(Kmp)$.

**Convergence** The objective function in (6) is a bi-convex problem for $\{(\boldsymbol{D}, \boldsymbol{P}), (\boldsymbol{A})\}$, e.g., by fixing $\boldsymbol{A}$ the function is convex for $\boldsymbol{D}$ and $\boldsymbol{P}$, and by fixing $\boldsymbol{D}$ and $\boldsymbol{P}$ the function is convex for $\boldsymbol{A}$. The convergence of such a problem has already been intensively studied [23], and the proposed optimization algorithm is actually an alternate convex search (ACS) algorithm. Since we have the optimal solutions of updating $\boldsymbol{A}$, $\boldsymbol{P}$ and $\boldsymbol{D}$, and our objective function has a general lower bound 0, our algorithm is guaranteed to converge to a stationary point. A detailed convergence analysis can be found in our supplementary file.

It is empirically found that the proposed DPL algorithm converges rapidly. Fig. 2 shows the convergence curve of our algorithm on the AR face dataset [24]. One can see that the energy drops quickly and becomes very small after 10 iterations. In most of our experiments, our algorithm will converge in less than 20 iterations.

# 3   Experimental Results

We evaluate the proposed DPL method on various visual classification datasets, including two face databases (Extended YaleB [22] and AR [24]), one object categorization database (Caltech101) [25], and one action recognition database (UCF 50 action [26]). These datasets are widely used in previous works [5, 9] to evaluate the DL algorithms.

Besides the classification accuracy, we also report the training and testing time of competing algorithms in the experiments. All the competing algorithms are implemented in Matlab except for SVM which is implemented in C. All experiments are run on a desktop PC with 3.5GHz Intel CPU and 8 GB memory. The testing time is calculated in terms of the average processing time to classify a single query sample.

## 3.1   Parameter setting

There are three parameters, $m$, $\lambda$ and $\tau$ in the proposed DPL model. To achieve the best performance, in face recognition and object recognition experiments, we set the number of dictionary atoms as its maximum (i.e., the number of training samples) for all competing DL algorithms, including the proposed DPL. In the action recognition experiment, since the samples per class is relatively big, we set the number of dictionary atoms of each class as 50 for all the DL algorithms. Parameter $\tau$ is an algorithm parameter, and the regularization parameter $\lambda$ is to control the discriminative property of $\boldsymbol{P}$. In all the experiments, we choose $\lambda$ and $\tau$ by 10-fold cross validation on each dataset. For all the competing methods, we tune their parameters for the best performance.

## 3.2   Competing methods

We compare the proposed DPL method with the following methods: the base-line nearest subspace classifier (NSC) and linear support vector machine (SVM), sparse representation based classification (SRC) [2] and collaborative representation based classification (CRC) [21], and the state-of-the-art DL algorithms DLSI [8], FDDL [9] and LC-KSVD [5]. The original DLSI represents the test sample by each class-specific sub-dictionary. The results in [9] have shown that by coding the test sample collaboratively over the whole dictionary, the classification performance can be greatly improved.

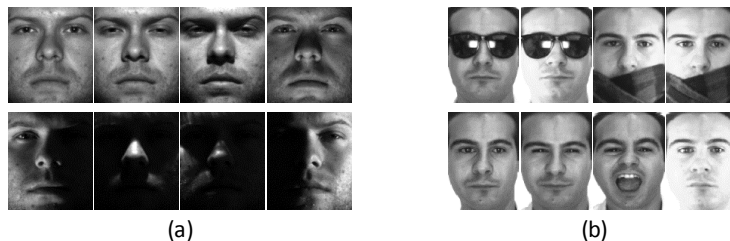

|   (a)   |   (b)   |

Figure 3: Sample images in the (a) Extended YaleB and (b) AR databases.

Therefore, we follow the use of DLDI in [9] and denote this method as DLSI(C). For the two variants of LC-KSVD proposed in [5], we adopt the LC-KSVD2 since it can always produce better classification accuracy.

### 3.3 Face recognition

We first evaluate our algorithm on two widely used face datasets: Extended YaleB [22] and AR [24]. The Extended YaleB database has large variations in illumination and expressions, as illustrated in Fig. 3(a). The AR database involves many variations such as illumination, expressions and sunglass and scarf occlusion, as illustrated in Fig. 3(b).

We follow the experimental settings in [5] for fair comparison with state-of-the-arts. A set of 2,414 face images of 38 persons are extracted from the Extended YaleB database. We randomly select half of the images per subject for training and the other half for testing. For the AR database, a set of 2,600 images of 50 female and 50 male subjects are extracted. 20 images of each subject are used for training and the remain 6 images are used for testing. We use the features provided by Jiang et al. [5] to represent the face image. The feature dimension is 504 for Extended YaleB and 540 for AR. The parameter $\tau$ is set to 0.05 on both the datasets and $\lambda$ is set to 3e-3 and 5e-3 on the Extended YaleB and AR datasets, respectively. In these two experiments, we also compare with the max-margin dictionary learning (MMDL) [10] algorithm, whose recognition accuracy is cropped from the original paper but the training/testing time is not available.

Table 1: Results on the Extended YaleB database.

|  | Accuracy (%) | Training time (s) | Testing time (s) |
|---|---|---|---|
| NSC | 94.7 | no need | 1.41e-3 |
| SVM | 95.6 | 0.70 | 3.49e-5 |
| CRC | 97.0 | no need | 1.92e-3 |
| SRC | 96.5 | no need | 2.16e-2 |
| DLSI(C) | 97.0 | 567.47 | 4.30e-2 |
| FDDL | 96.7 | 6,574.6 | 1.43 |
| LC-KSVD | 96.7 | 412.58 | 4.22e-4 |
| MMDL | 97.3 | - | - |
| DPL | **97.5** | 4.38 | 1.71e-4 |

Table 2: Results on the AR database.

|  | Accuracy (%) | Training time (s) | Testing time (s) |
|---|---|---|---|
| NSC | 92.0 | no need | 3.29e-3 |
| SVM | 96.5 | 3.42 | 6.16e-5 |
| CRC | 98.0 | no need | 5.08e-3 |
| SRC | 97.5 | no need | 3.42e-2 |
| DLSI(C) | 97.5 | 2,470.5 | 0.16 |
| FDDL | 97.5 | 61,709 | 2.50 |
| LC-KSVD | 97.8 | 1,806.3 | 7.72e-4 |
| MMDL | 97.3 | - | - |
| DPL | **98.3** | 11.30 | 3.93e-4 |

**Extended YaleB database** The recognition accuracies and training/testing time by different algorithms on the Extended YaleB database are summarized in Table 1. The proposed DPL algorithm achieves the best accuracy, which is slightly higher than MMDL, DLSI(C), LC-KSVD and FDDL. However, DPL has obvious advantage in efficiency over the other DL algorithms.

**AR database** The recognition accuracies and running time on the AR database are shown in Table 2. DPL achieves the best results among all the competing algorithms. Compared with the experiment on Extended YaleB, in this experiment there are more training samples and the feature dimension is higher, and DPL′s advantage of efficiency is much more obvious. In training, it is more than 159 times faster than DLSI and LC-KSVD, and 5,460 times faster than FDDL.

### 3.4 Object recognition

In this section we test DPL on object categorization by using the Caltech101 database [25]. The Caltech101 database [25] includes 9,144 images from 102 classes (101 common object classes and a background class). The number of samples in each category varies from 31 to 800. Following the experimental settings in [5, 27], 30 samples per category are used for training and the rest are

Table 3: Recognition accuracy(%) & running time(s) on the Caltech101 database.

|         | Accuracy | Training time | Testing time |
|---------|----------|---------------|--------------|
| NSC     | 70.1     | no need       | 1.79e-2      |
| SVM     | 64.6     | 14.6          | 1.81e-4      |
| CRC     | 68.2     | no need       | 1.38e-2      |
| SRC     | 70.7     | no need       | 1.09         |
| DLSI(C) | 73.1     | 97,200        | 1.46         |
| FDDL    | 73.2     | 104,000       | 12.86        |
| LC-KSVD | 73.6     | 12,700        | 4.17e-3      |
| DPL     | **73.9** | 134.6         | 1.29e-3      |

used for testing. We use the standard bag-of-words (BOW) + spatial pyramid matching (SPM) framework [27] for feature extraction. Dense SIFT descriptors are extracted on three grids of sizes $1 \times 1$, $2 \times 2$, and $4 \times 4$ to calculate the SPM features. For a fair comparison with [5], we use the vector quantization based coding method to extract the mid-level features and use the standard max pooling approach to build up the high dimension pooled features. Finally, the original 21,504 dimensional data is reduced to 3,000 dimension by PCA. The parameters $\tau$ and $\lambda$ used in our algorithm are 0.05 and 1e-4, respectively.

The experimental results are listed in Table 3. Again, DPL achieves the best performance. Though its classification accuracy is only slightly better than the DL methods, its advantage in terms of training/testing time is huge.

## 3.5  Action recognition

Action recognition is an important yet very challenging task and it has been attracting great research interests in recent years. We test our algorithm on the UCF 50 action database [26], which includes 50 categories of 6,680 human action videos from YouTube. We use the action bank features [28] and five-fold data splitting to evaluate our algorithm. For all the comparison methods, the feature dimension is reduced to 5,000 by PCA. The parameters $\tau$ and $\lambda$ used in our algorithm are both 0.01.

The results by different methods are reported in Table 4. Our DPL algorithm achieves much higher accuracy than its competitors. FDDL has the second highest accuracy; however, it is 1,666 times slower than DPL in training and 83,317 times slower than DPL in testing.

Table 4: Recognition accuracy(%) & running time(s) on the UCF50 action database

| Methods | Accuracy | Training time | Testing time |
|---------|----------|---------------|--------------|
| NSC     | 51.8     | no need       | 6.11e-2      |
| SVM     | 57.9     | 59.8          | 5.02e-4      |
| CRC     | 60.3     | no need       | 6.76e-2      |
| SRC     | 59.6     | no need       | 8.92         |
| DLSI(C) | 60.0     | 397,000       | 10.11        |
| FDDL    | 61.1     | 415,000       | 89.15        |
| LC-KSVD | 53.6     | 9,272.4       | 0.12         |
| DPL     | **62.9** | 249.0         | 1.07e-3      |

## 4  Conclusion

We proposed a novel projective dictionary pair learning (DPL) model for pattern classification tasks. Different from conventional dictionary learning (DL) methods, which learn a single synthesis dictionary, DPL learns jointly a synthesis dictionary and an analysis dictionary. Such a pair of dictionaries work together to perform representation and discrimination simultaneously. Compared with previous DL methods, DPL employs projective coding, which largely reduces the computational burden in learning and testing. Performance evaluation was conducted on publically accessible visual classification datasets. DPL exhibits highly competitive classification accuracy with state-of-the-art DL methods, while it shows significantly higher efficiency, e.g., hundreds to thousands times faster than LC-KSVD and FDDL in training and testing.

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
