[Supplementary Material]

# Supplementary materials to "Projective dictionary pair learning for pattern classification"

**Shuhang Gu**[1], **Lei Zhang**[1], **Wangmeng Zuo**[2], **Xiangchu Feng**[3]
[1]Dept. of Computing, The Hong Kong Polytechnic University, Hong Kong, China
[2]School of Computer Science and Technology, Harbin Institute of Technology, Harbin, China
[3]Dept. of Applied Mathematics, Xidian University, Xi′an, China
{cssgu, cslzhang}@comp.polyu.edu.hk
cswmzuo@gmail.com, xcfeng@mail.xidian.edu.cn

In this file, we provide a detailed convergence analysis for the proposed DPL algorithm. We first introduce some related definitions of the bi-convex optimization problem [1, 2], and present the conclusions of the convergence of the alternative convex search (ACS) algorithm [3]. Then, we analyze the bi-convex property of our objective function, and show the equivalence between the ACS algorithm and our optimization algorithm.

## 1    Bi-convex problem and alternative convex search algorithm

We first introduce the basic definitions of bi-convex optimization problem, and then introduce the ACS algorithm used to solve it. The convergence of the ACS algorithm has been analyzed in [2].

**Definition 1.** *(Definition 1.1, [2]) The set $B \subseteq X \times Y$ is called a bi-convex set on $X \times Y$, if $B_x := \{y \in Y : (x, y) \in B\}$ is convex for every $x \in X$ and $B_y := \{x \in X : (x, y) \in B\}$ is convex for every $y \in Y$.*

**Definition 2.** *(Definition 1.2, [2]) A function $f : B \to \Re$ on a bi-convex set $B \subseteq X \times Y$ is called a bi-convex function on $B$, if*

$$f_x(\bullet) := f(x, \bullet) : B_x \to \Re \text{ is a convex function on } B_x \text{ for every fixed } x \in X,$$

$$f_y(\bullet) := f(\bullet, y) : B_y \to \Re \text{ is a convex function on } B_y \text{ for every fixed } y \in Y.$$

**Definition 3.** *(Definition 1.3, [2]) An optimization problem of the form*

$$min_{x,y}\{f(x, y) : (x, y) \in B\} \tag{1}$$

*is said to be a bi-convex optimization problem if the feasible set $B$ is bi-convex on $X \times Y$ and the objective function $f$ is bi-convex on $B$.*

**Definition 4.** *(Definition 4.1, [2]) Let $f : B \to \Re$ be a given function and let $(x^*, y^*) \in B$. Then, $(x^*, y^*)$ is called a partial optimum of $f$ on $B$, if*

$$f(x^*, y^*) \leq f(x, y^*), \ \forall x \in B_{y^*}; \ and \ f(x^*, y^*) \leq f(x^*, y), \ \forall y \in B_{x^*}. \tag{2}$$

Please note that the condition of partial optimum is weak and the condition in **Definition 4** is not sufficient to guarantee a local optimum.

The ACS algorithm [3] is widely used to solve the bi-convex optimization problem. ACS is a special case of the block relaxation methods [4], and the relationship between ACS and other algorithms can be found in the survey paper [2]. For a general bi-convex optimization problem defined in (1), the ACS algorithm chooses an arbitrary starting point $z_0 = (x_0, y_0) \in B$ and alternatively solves the problem as follows.

**Step 1:** Fix $y_i$ and solve the following convex optimization problem:

$$\min_x\{f(x, y_i), x \in B_{y_i}\}. \tag{3}$$

If there exists an optimal solution $x^* \in B_{y_i}$ to (3), set $x_{i+1} = x^*$; otherwise, STOP.

**Step 2:** Fix $x_{i+1}$ and solve the following convex optimization problem:

$$\min_y \{f(x_{i+1}, y), y \in B_{x_{i+1}}\}. \tag{4}$$

If there exists an optimal solution $y^* \in B_{x_{i+1}}$ to (4), set $y_{i+1} = y^*$; otherwise, STOP.

**Step 3:** If the stopping criterion is satisfied, we have the solution $z_{i+1} = (x_{i+1}, y_{i+1})$; otherwise, go back to **Step 1**.

The convergence of ACS algorithm has been intensively studied in [2]. In this file, we only list two theorems which are related to the convergence analysis of the proposed DPL method. For the proof of the theorems, please refer to [2].

**Theorem 1.** *(Theorem 4.5, [2]) Let $B \subseteq \Re^n \times \Re^m$, let $f : B \to \Re$ be bounded from below, and let the optimization problems (3) and (4) be solvable. Then the sequence $\{f(z_i)\}_{i \in N}$ generated by ACS converges monotonically.*

Theorem **1** provides a convergence guarantee for the energy of the objective function $\{f(z_i)\}_{i \in N}$. However, the convergence of $\{f(z_i)\}_{i \in N}$ can not ensure the convergence of variable $\{z_i\}_{i \in N}$. To further analyze the convergence of the variables, we need another theorem.

**Theorem 2.** *(Theorem 4.9, [2]) Let $X \subseteq \Re^n$ and $Y \subseteq \Re^m$ be closed sets and let $f : X \times Y \to \Re$ be continuous. Let the optimization problems (3) and (4) be solvable.*

*1. If the sequence $\{z_i\}_{i \in N}$ generated by ACS algorithm is contained in a compact set, then the sequence has at least one accumulation point.*

*2. Suppose that for each accumulation point $z^* = (x^*, y^*)$ of the sequence $\{z_i\}_{i \in N}$ the optimal solution of (3) with $y = y^*$ or the optimal solution of (4) with $x = x^*$ is unique, then all accumulation points are partial optima and have the same function value.*

*3. If for each accumulation point $z^* = (x^*, y^*)$ of the sequence $\{z_i\}_{i \in N}$ the optimal solutions of both (3) with $y = y^*$ and (4) with $x = x^*$ are unique, then*

$$\lim_{i \to \infty} \|z_{i+1} - z_i\| = 0, \tag{5}$$

*and the accumulation points form a compact continuum C.*

## 2  Convergence analysis of DPL

We analyze the property of the proposed DPL model and present some remarks on the convergence of our optimization algorithm .

**Remark 1.** *The sequence of $\{f(\mathbf{D}^i, \mathbf{P}^i, \mathbf{A}^i)\}_{i \in N}$ generated by our DPL algorithm converges monotonically.*

*Proof.* In the proposed DPL model, three variables $\boldsymbol{D}$, $\boldsymbol{P}$ and $\boldsymbol{A}$ are optimized:

$$\{\boldsymbol{P}^*, \boldsymbol{A}^*, \boldsymbol{D}^*\} = \arg \min_{\boldsymbol{P}, \boldsymbol{A}, \boldsymbol{D}} \sum_{k=1}^{K} \|X_i - D_k A_k\|_F^2 + \tau \|P_k X_k - A_k\|_F^2 + \lambda \|P_k \bar{X}_k\|_F^2, \ \ s.t. \ \|d_i\|_2^2 \le 1. \tag{6}$$

By fixing $\boldsymbol{A}$, the variables $\boldsymbol{D}$ and $\boldsymbol{P}$ are separable, and they can be termed as a single variable. Thus, the optimization problem in (6) is a bi-convex problem of $\{\boldsymbol{A}, (\boldsymbol{D}, \boldsymbol{P})\}$. In our DPL training algorithm, we alternatively solve the following two convex optimization problems:

$$\begin{cases} \boldsymbol{A}^* = \arg \min_{\boldsymbol{A}} \sum_{k=1}^{K} \|X_k - D_k A_k\|_F^2 + \tau \|P_k X_k - A_k\|_F^2, & (7) \\[2mm] \{\boldsymbol{D}^*, \boldsymbol{P}^*\} = \arg \min_{\boldsymbol{D}, \boldsymbol{P}} \sum_{k=1}^{K} \|X_k - D_k A_k\|_F^2 + \tau \|P_k X_k - A_k\|_F^2 + \lambda \|P_k \bar{X}_k\|_F^2, & s.t. \ \|d_i\|_2^2 \le 1. (8) \end{cases}$$

The continuous and differentiable functions (7) and (8) correspond to (3) and (4) in the ACS algorithm, and the whole objective function (6) has a general lower bound 0. Thus, based on **Theorem 1**, the optimization procedure of our DPL algorithm are guaranteed to converge monotonically in terms of energy. □

**Remark 2.** *The sequence of $\{\mathbf{D}^i, \mathbf{P}^i, \mathbf{A}^i\}_{i \in N}$ generated by our DPL algorithm has at least one accumulation point. All the accumulation points are partial optima of $f$ and have the same function value.*

*Proof.* The variables $\boldsymbol{X}$ and $\boldsymbol{D}$ in (6) have normalized columns. One can easily find that the objective function satisfies $f(\boldsymbol{P}, \boldsymbol{D}, \boldsymbol{A}) \to \infty$ if $\|\boldsymbol{P}\|_F \to \infty$ or $\|\boldsymbol{A}\|_F \to \infty$, which implies that the sequence $\{\boldsymbol{D}^i, \boldsymbol{P}^i, \boldsymbol{A}^i\}_{i \in N}$ generated by the DPL algorithm is bounded in a finite dimensional space. Thus, the compact set condition in **Theorem 1** is satisfied and the sequence has at least one accumulation point.

Furthermore, for any $\tau > 0$, the $\boldsymbol{A}$ subproblem in (7) is a strict convex problem and has a unique solution. The second condition in **Theorem 2** is satisfied, which ensures that all the accumulation points are partial optima and have the same function value. $\qquad\square$

**Remark 3.** *If the problem in (8) has a unique solution, then the sequence $\{\mathbf{D}^i, \mathbf{P}^i, \mathbf{A}^i\}_{i \in N}$ generated by our DPL algorithm satisfies:*

$$\lim_{i \to \infty} \|\mathbf{P}^{i+1} - \mathbf{P}^i\| + \|\mathbf{D}^{i+1} - \mathbf{D}^i\| + \|\mathbf{A}^{i+1} - \mathbf{A}^i\| = 0. \tag{9}$$

*Proof.* Based on **Remark 2**, the conditions 1 and 2 in **Theorem 2** are satisfied in our DPL algorithm. If we have the unique optimal solution of $(\boldsymbol{D}, \boldsymbol{P})$, we can have the conclusion (9) based on condition 3 in **Theorem 2**. $\qquad\square$

For the optimization problem in (8), the condition in **Remark 3** becomes that the class-specific matrices $\boldsymbol{A}_k \boldsymbol{A}_k^T$ and $\boldsymbol{X}_k \boldsymbol{X}_k^T + \bar{\boldsymbol{X}}_k \bar{\boldsymbol{X}}_k^T$ should be non-singular matrices, which is not satisfied in general in our DPL model. In many applications, the dimension of feature space is much higher than the sample number of each class, and $\boldsymbol{X}_k \boldsymbol{X}_k^T + \bar{\boldsymbol{X}}_k \bar{\boldsymbol{X}}_k^T$ is a singular matrix. So we add a small regularization term $\gamma \boldsymbol{I}$ in the implementation to ensure that we can get the optimum in each $\boldsymbol{P}$ subproblem (please refer to equation (10) in the main paper). As for the matrix $\boldsymbol{A}_k \boldsymbol{A}_k^T$, since in most cases the atom number of class-specific dictionary is less than the sample number of each class, $\boldsymbol{A}_k$ is a row full rank matrix, and thus $\boldsymbol{A}_k \boldsymbol{A}_k^T$ is non-singular.

Please note that even the condition in **Remark 3** is satisfied, the conclusion in **Remark 3** can only ensure that the change of the variables in adjacent iterations tends to be zero. There is no guarantee that the sequence $\{\boldsymbol{D}^i, \boldsymbol{P}^i, \boldsymbol{A}^i\}_{i \in N}$ will converge to a local minimum. However, in most practical applications, it is enough to terminate the iteration if the change of variables is less than a small threshold.