[Reviews · NeurIPS 2014]

Submitted by Assigned_Reviewer_25

The paper proposes projective dictionary pair learning (DPL) to jointly learn a synthesis and analysis dictionary so as to apply the dictionary learning approach to the classification setting while also enabling more efficient computation. Overall the paper proposes an interesting method. Even though the proposed method is a straight-forward extension of DL, the empirical results (esp the computational gains) looks quite promising.

It would have been great to see:
i) an accompanying theoretical analysis esp around convergence
ii) analysis of the complexity of method as sample size increases (and hence with increasing m)

It is interesting how the method imposes sparsity indirectly (as seen in Sec 2.4) by training $p^*_k$ to produce small coefficients. It would be interesting to see a discussion on the impact of this indirect sparsity imposition against conventional regularization.

Finally, there are a few typos in the paper that should be fixed (easily identifiable upon a re-reading).
Summary: Interesting method that allows for joint learning of analysis and synthesis dictionaries enabling a margin-sparsity tradeoff type setup well suited for classification problems. Theoretical analysis esp around convergence and some additional evaluation would have made the manuscript much stronger.

Submitted by Assigned_Reviewer_36

This paper proposes a discriminative dictionary learning framework for both synthesis and analysis dictionaries, namely projective dictionary pair learning (DPL), where the analysis dictionary is learned by linear projection. The purpose of the method is to improve the computational efficiency of the coding process by discarding the explicit optimization of sparsity regularizers, e.g. l1/l0 norm without sacrificing the classification accuracy.

While the paper is easy to follow there are several typos/grammatical errors/incorrect equation references, e.g. on page 4, (8) in the last sentence should be (9), that needs to be corrected. The paper deals with an topic that is of significant interest. Nevertheless, the proposed method has marginal novelty because there are related methods already existing in the literature. For instance, see the following missing but very related reference:

Zhizhao Feng*, Meng Yang*, Lei Zhang, Yan Liu, and David Zhang, Joint discriminative dimensionality reduction and dictionary learning for face recognition. To Appear in Pattern Recognition. (* contribute equally to this work).

In addition the reviewer has serious concerns about the proposed approach:

(1) Line 191-192, bi-linear optimization problem of the objective function in Eq. (5). Why is it bi-linear?

(2) Convergence. Even though the authors show an example of convergence in practice, there is no guarantee because of the normalization of D. Consequently, the convergence of the method is highly dependent on the data, limiting the scalability of the method. Is it possible to analyze the convergence theoretically?

(3) Classification. The proposed algorithm optimizes the variables D, P, and A alternatively in Eq (6). However, the proposed classification rule is based on Eq (5). Why don't you still use Eq (6) as your classification rule? That is, first estimate A for a test sample using Eq (8), and then checking the reconstruction error for each class as proposed. To me this is more consistent with what you optimize, and the computational cost is the same as the proposed one if pre-computing some matrices in Eq (8).

(4) Line 306, robustness to variable m. I have my doubts about this statement. My line of thinking is as follows: This parameter essentially determines the rank of the matrix DP in Eq (5) explicitly. This actually is equivalent to controlling the model complexity by essentially enforcing low-rank constraints. If we carefully check the details in the experimental section, we can find that the value of this parameter is always much smaller than the dimensionality of features used in each dataset, which guarantees the low-rank property. So why don't you plot a figure that contains the classification accuracy with increasing m?
Summary: While the topic of the paper is of interest to the NIPS community the proposed method is incremental with respect to existing body of work in the literature.

Submitted by Assigned_Reviewer_40

This paper proposes a new discriminative dictionary learning (DL) method where both a synthesis dictionary and an analysis dictionary are jointly learned: the class-dependent synthesis dictionary reconstructs the data as in conventional DL methods, and at the same time, the reconstructing codes can be obtained as linear projections of data using a class-dependent analysis dictionary. One benefit of this new approach is getting rid of the computationally expensive L-1 or L-0 sparsity penalty on the code.

Below are ratings (1~5, from low to high) for Quality, Clarity, Originality and Significance.

1) Quality score (1~5): 4. The idea of jointly learning a synthesis dictionary (for reconstructing the data) and an analysis dictionary (for projecting the code) is interesting. The formula is well thought out. It's a very smart trick to replace the computationally expensive sparsity penalty by the frobenius-norm panelty, which promotes the idea that "cross-class projection should land on a nearly null space, i.e., zero code". Empirical studies are convincing with many state-of-the-art DL methods. The proposed method provides computationally efficient solution while maintain top classification accuracy.

2) Clarity score (1~5): 4. The paper is clearly written for most parts. Some typos: Line 213, (8) should be (9)? Line 236, (8) should be (9)?

3) Originality (1~5): 3. As mentioned above, the idea of jointly learning a synthesis dictionary (for reconstructing the data) and an analysis dictionary (for projecting the code) are interesting.

4) Significance (1~5): 4. This paper could possibly impact future work on discriminative dictionary learning.

Summary: The idea of jointly learning a synthesis dictionary (for reconstructing the data) and an analysis dictionary (for projecting the code) is interesting. The formula that gets rid of sparsity constraint is well thought out. Empirical studies are sufficient.
Author Feedback
Author rebuttal: We thank all the reviewers for their constructive comments. R_25 and R_40 recognized the novelty and performance of our method. R_36 argued that our work is much related to a paper published in Patter Recognition. Actually, our work is distinctly different from the PR paper. The motivations, models and algorithms are all different. Our work is the first to learn a discriminative dictionary pair for pattern classification. It has much faster training and testing speed than state-of-the-art but with very competitive accuracy. We sincerely ask R_36 to double check the two papers and render a more justified recommendation.

What below are our itemized responses to the reviewers’ comments.

Reviewer_25:

Q1. Theoretical analysis on convergence

Our algorithm can converge to a stationary point. The objective function in (6) is bi-convex for {(D,P), (A)}. It is convex in A for fixed (D,P). It is also convex in (D,P) for fixed A since, fortunately, the subproblem on (D,P) in our model is separable for D and P (please refer to (9)). There are many papers published on bi-convex optimization. We adopted an alternate convex search algorithm like in [15] and [R1]. It iterates between solving the subproblem on A by fixing (D,P) and solving the subproblem on (D,P) by fixing A. Note that (6) has a lower bound 0. According to Theorem 4.7 in [R1], our algorithm is guaranteed to converge to a stationary point. We will add the theoretical analysis of convergence in the revision.

[R1] J. Gorski, F. Pfeuffer, and K. Klamroth, “Bicovex sets and optimization with biconvex functions – a survey and extensions,” Mathematical Methods of Operations Research 66, no. 3 (2007): 373-407.

Q2. Complexity

In Section 2.5, we’ve analyzed the complexity of our algorithm with n (the number of training samples) and m (the number of atoms in the dictionary). In training, the complexity w.r.t. n is O(n), and the complexity w.r.t. m is O(m^3). One of the motivations of discriminative dictionary learning is to learn a more compact set of atoms to replace the training set for efficient sample representation. In practice, m is smaller than n, especially when n is big.

Q3. Sparsity

Thanks for the suggestion. This is really an interesting problem. No doubt sparsity helps pattern classification. Conventional methods use L_p (p <= 1) or L_p,q norm regularization to impose sparsity on the coefficients. However, the drawback is the high complexity due to the nonlinear sparse coding process. In this paper, we train linear projections $P^*_k$ to produce large coefficients on class k but small coefficients on other classes. As shown in Fig. 1, this actually results in a kind of group-sparsity on the coefficients. Compared with the conventional L_p,q norm induced (group) sparsity, the sparsity induced by $P^*_k$ may be weaker; however, they can still lead to very competitive classification accuracy while reducing significantly the complexity. We will add more discussions in the revision.

Q4. Typos

We will correct the typos in revision and proof read carefully the manuscript.

Reviewer_36:

Q1. Difference from the PR paper

Our work is distinctly different from the PR paper mentioned by this reviewer.

First of all, the motivations are very different. The P learned in the PR paper is a dimensionality reduction matrix. It is to reduce the dimension of data in X, and PX is then coded over D to obtain the sparse codes. In our paper, the learned P is not for dimensionality reduction but to produce the codes directly via PX.

Second, the learning models are totally different. The data term in the PR paper is ||PX-DA||, while that in our model is ||X-DPX||. Moreover, the regularizers on D and P in the two models are very different. Particularly, the L_1 norm regularizer on coding coefficients is needed in the PR paper, but no L_1 regularizer is involved in our model.

Consequently, the learning algorithms are different. There are closed-form solutions of P, D and A in each iteration in our algorithm, which is much simpler and faster than the PR paper.

Clearly, our paper has little similarity and overlap with the PR paper. To the best of our knowledge, it is the first work to learn a pair of discriminative dictionaries for pattern classification.

Q2. Bi-linear

Sorry that we made a typo. It should be “bi-convex”.

Q3. Convergence

Yes the convergence can be analyzed theoretically. Please refer to our response to Q1 of Reviewer_25.

Q4. Classification rule

Matrix A is introduced to guide the learning of D and P. In the training stage, we know the class labels of sample matrix X. By enforcing group sparsity on A, Eq. (8) can be derived to update A. However, in the testing stage we don’t know the class label of query sample y, and using Eq. (8) to compute the coding coefficient of y is not suitable and this cannot fully exploit the discrimination capability of P_k. Therefore, we use P_k*y to generate the code for classification.

Q5. Robustness to m

The selection of m depends on n (the number of training samples). In dictionary learning, it is commonly set m<=n. In visual classification, the feature dimensionality p is usually much higher than n (i.e., the so called small sample problem), and hence m is much smaller than p. Our empirical results show that when increasing m from a very small number, the recognition rate could be improved; however, with the increase of m, the improvement becomes very small. Our method is not sensitive to m in a wide range. For example, in the experiment on Extended Yale B, the classification rates (%) with m (n=30) are:

m=1: 91.65
m=3: 95.66
m=5: 96.99
m=7: 97.41
m=9: 97.58
m=10: 97.58
m=15: 97.66
m=20: 97.66
m=25: 97.66
m=30: 97.66

One can see that when m >= 7, the rates are nearly the same.

Reviewer_40:

Thanks for the comments. We will correct the typos in the revision.